# Green Synthesis of Endolichenic Fungi Functionalized Silver Nanoparticles: The Role in Antimicrobial, Anti-Cancer, and Mosquitocidal Activities

**DOI:** 10.3390/ijms231810626

**Published:** 2022-09-13

**Authors:** Yugal Kishore Mohanta, Debasis Nayak, Awdhesh Kumar Mishra, Ishani Chakrabartty, Manjit Kumar Ray, Tapan Kumar Mohanta, Kumananda Tayung, Rajapandian Rajaganesh, Murugan Vasanthakumaran, Saravanan Muthupandian, Kadarkarai Murugan, Gouridutta Sharma, Hans-Uwe Dahms, Jiang-Shiou Hwang

**Affiliations:** 1Department of Applied Biology, School of Biological Sciences, University of Science and Technology Meghalaya, Ri-Bhoi 793101, Meghalaya, India; 2Department of Wildlife and Biodiversity Conservation, Maharaja Sriram Chandra Bhanj Deo University, Baripada 757003, Odisha, India; 3Department of Biotechnology, Yeungnam University, Gyeongsan 38541, Korea; 4Natural and Medical Sciences Research Centre, University of Nizwa, Nizwa 616, Oman; 5Department of Botany, Gauhati University, Jalukbari, Guwahati 781014, Assam, India; 6Department of Zoology, Bharathiar University, Coimbatore 641046, Tamil Nadu, India; 7AMR and Nanotherapeutics Laboratory, Department of Pharmacology, Saveetha Dental College, Saveetha Institute of Medical and Technical Sciences (SIMATS), Chennai 600077, Chennai, India; 8Department of Biomedical Science and Environmental Biology, Kaohsiung Medical University (KMU), Kaohsiung 80708, Taiwan; 9Department of Marine Biotechnology and Resources, National Sun Yat-sen University, Kaohsiung 80424, Taiwan; 10Research Center for Environmental Medicine, Kaohsiung Medical University (KMU), Kaohsiung 80708, Taiwan; 11Institute of Marine Biology, National Taiwan Ocean University, Keelung 20224, Taiwan; 12Center of Excellence for Ocean Engineering, National Taiwan Ocean University, Keelung 20224, Taiwan; 13Center of Excellence for the Oceans, National Taiwan Ocean University, Keelung 20224, Taiwan

**Keywords:** endolichenic fungi, *Talaromyces funiculosus*, antibacterial, antioxidant, anti-proliferative activity, biological mosquito control

## Abstract

Green nanotechnology is currently a very crucial and indispensable technology for handling diverse problems regarding the living planet. The concoction of reactive oxygen species (ROS) and biologically synthesized silver nanoparticles (AgNPs) has opened new insights in cancer therapy. The current investigation caters to the concept of the involvement of a novel eco-friendly avenue to produce AgNPs employing the wild endolichenic fungus *Talaromyces funiculosus.* The synthesized *Talaromyces funiculosus*–AgNPs were evaluated with the aid of UV visible spectroscopy, dynamic light scattering (DLS), Fourier infrared spectroscopy (ATR-FTIR), X-ray diffraction (XRD), scanning electron microscopy (SEM), and transmission electron microscopy (TEM). The synthesized *Talaromyces funiculosus*–AgNPs (*TF*-AgNPs) exhibited hemo-compatibility as evidenced by a hemolytic assay. Further, they were evaluated for their efficacy against foodborne pathogens *Staphylococcus aureus, Streptococcus faecalis*, *Listeria innocua*, and *Micrococcus luteus* and nosocomial *Pseudomonas aeruginosa, Escherichia coli, Vibrio cholerae,* and *Bacillus subtilis* bacterial strains. The synthesized *TF*-AgNPs displayed cytotoxicity in a dose-dependent manner against MDA-MB-231 breast carcinoma cells and eventually condensed the chromatin material observed through the Hoechst 33342 stain. Subsequent analysis using flow cytometry and fluorescence microscopy provided the inference of a possible role of intracellular ROS (OH^−^, O^−^, H_2_O_2_, and O_2_^−^) radicals in the destruction of mitochondria, DNA machinery, the nucleus, and overall damage of the cellular machinery of breast cancerous cells. The combined effect of predation by the cyclopoid copepod *Mesocyclops aspericornis* and TF-AgNPS for the larval management of dengue vectors were provided. A promising larval control was evident after the conjunction of both predatory organisms and bio-fabricated nanoparticles. Thus, this study provides a novel, cost-effective, extracellular approach of *TF*-AgNPs production with hemo-compatible, antioxidant, and antimicrobial efficacy against both human and foodborne pathogens with cytotoxicity (dose dependent) towards MDA-MB-231 breast carcinoma.

## 1. Introduction

Green nanotechnology has endorsed intense approaches with diverse forms, such as nanoscale material synthesis (biological or green methods), and made use of their peculiar physicochemical and optoelectronic properties [1,2]. Recent advancements in our understanding of self-cluster mechanisms has led to unique approaches for the assembly of nanoparticles into predetermined superstructures, thus establishing that green nanotechnology will provide a progressive, important contribution towards profuse essential technologies in the current era of development [3,4]. Commercial utilization of nanoparticles requires their accumulation and packaging in thin-film form, and in the upgrade of the concept, the “bottom-up” approach has been gathering substantial attention [5,6]. However, there is a need to promote uncluttered, nontoxic, and environmentally friendly processes, i.e., the green chemistry process for synthesis and aggregation of nanoparticles, which has forced the scientific world to thoroughly consider biological schemes [7,8]. Hence, this has led to the improvement in the biomimetic passage for the development of advanced nanomaterials. There is no scarcity of examples of biogenic synthesis of nanomaterials from different plants or the animal kingdom, or unicellular organisms that yield inorganic nanomaterials either from intra- or extracellular approaches [9]. Very well-known examples involved in nanomaterial synthesis include magneto-tactic bacteria (magnetic nanoparticles) [10], diatoms (siliceous materials) [11] and S-layer bacteria (gypsum and calcium carbonate layers) [12,13].

For centuries, humans and pathogens have been actively involved in the escalation of a “health war” and large numbers of mortalities were caused by pathogenic microorganisms. With the discovery of “penicillin” as a pioneer antibiotic, most pathogenic bacteria could be defeated [14]. Meanwhile, microbes have emerged and developed strong resistance [15]. As a result, research is required to explore alternative antibiotics [16,17]. To accomplish this intention, researchers have discovered that the silver ion (Ag^+^) has the most vigorous antimicrobial capabilities through an expanded influence on pathogenic microorganisms [18].

Cancer has evolved as the second leading cause of death. Reports in 2018 revealed 18 million new cases and approximately 9.5 million deaths annually worldwide. Analysis of the updating clinical research data reveals an expected 29.5 million new cases and 16.4 million deaths by 2040 [19,20]. Breast cancer is now the most frequent cancer in women and considered as the second most common cancer overall, and there were 2.3 million diagnoses and 685,000 lakhs deaths globally in 2020. In total, 7.8 million women were under diagnosis making it the world’s most ubiquitous cancer [21]. The treatment of cancer cells using existing methods such as surgery, chemotherapy, and radiotherapy, is highly toxic to normal cells and is considered redundant as well as inefficient [22,23,24]. In order to scale down the complications due to chemotherapy, the development of exclusive anti-cancer drugs combined with nanocomposites was explored [25]. The anticancer activity of AgNPs is proven through several reports [26,27,28,29,30]. In order to avoid the cytotoxic effects of AgNPs (chemically synthesized) on normal human cells, the biosynthesized AgNPs are in the spotlight of researchers at present.

Fungi have an admirable future regarding the synthesis of many unique natural compounds intracellularly or extracellularly that can be utilized in diverse applications. Around 6400 potential natural bioactive entities are synthesized or produced by filamentous (microscopic) fungi (imperfect fungi and ascomycetes) and many other fungi [31]. Novel chemicals from these microorganisms are universally exploited as both reducing as well as stabilizing agents. They show a potential in heavy-metal resistance and the scope of incarnating and bio-accumulating metals. However, the fungi can be efficiently cultured on a large scale (“nano-factories”) and used to harvest nanoparticles of different shapes and sizes [32,33]. It is noteworthy that fungi are important resources over other microbial resources, as they can yield bulky enzymes (proteins); few of them can be utilized for the efficient and green synthesis of metal and non-metal nanoparticles [34,35]. The extracellular synthesis method is highly preferable as it is an inexpensive and less-time-consuming procedure with low-cost experimental procedures [32,33,36,37]. However, synthesized nanoparticles that are present in a dispersive form should be elutriated to remove the fungal cell residues and scum, which can be enacted using different filtration techniques such as gel/membrane filtration, dialysis, or ultracentrifugation [38,39]. Several reports have been found on the synthesis of nanoparticles using different fungal species but only a few studies used endolichenic fungi during times of green myco-synthesis of nanoparticles.

Mosquitoes are vector organisms and are spreading many important diseases such as malaria, dengue fever, filariasis, yellow fever, Japanese encephalitis, West Nile, and Zika virus [40]. Efficient vector management is of paramount importance [41]. A synthetic chemical poses environmental toxicity and affects natural enemy complexes in aquatic ecosystems and, hence, provides an alternate source for vector management programs [42].

Herein, we studied the green synthesis of AgNPs using the endolichenic fungus *Talaromyces funiculosus* and its potential applications in the food industry and the biomedical industry through the evaluation of antimicrobial activity against foodborne and human pathogens and antioxidant activity and cytotoxic activity against epidermal cell lines (HaCaT) and breast adenocarcinoma (MDA-MB 231). In addition, the results on the use of copepods *M. aspericornis* as efficient predators for mosquito larval control and their safety as non-target organisms after using silver nanoparticles were evaluated. The current work served as a potential platform for the synthesis of AgNPs without the association of toxic chemicals and physical radiation. 

## 2. Results and Discussion

### 2.1. Synthesis and Characterization of TF-AgNPs

UV spectroscopy provides a primary inference for the synthesis of fungus-mediated AgNPs. Figure 1 shows the UV spectral analysis of the synthesized AgNPs from the endolichenic fungi *T. funiculosus.* The presence of an absorbance peak at around 430 nm shows that the synthesized nanoparticles had an optimum size range as previously demonstrated by [43,44]. The reduction of Ag^2+^ to Ag^0^, coupled with the excitation of surface plasmon resonance (SPR), results in this peak. Various studies have also reported similar absorption peaks for the green-synthesized AgNPs mediated by fungal biomass. Hence, a primary optical screening test suggested the synthesis of AgNPs where after incubation the fungus readily reduced silver nitrate into Ag^+^ ions corresponding to the dark brown color in the conical flask. Generally, a sharp peak at a lower wavelength is indicative of the small particle size of NPs, whereas a flat peak at a higher wavelength means large-sized particles. Increasing the fungal biomass concentration (and subsequently reduction agent) results in an increase in NP size and leads to shifting of the SPR peak towards a higher wavelength. The increase in concentration of fungal biomass allows stabilization of large-sized NPs as it acts as a surface functionalizing ligand [45]; this is also reflected by the color change of the *TF*-AgNPs solution from golden yellow to dark brown. Therefore, at a higher concentration, large-sized NPs with low aggregation were formed. However, with the increase in fungal biomass, the intensities of the SPR peaks decreased, which is suggestive of a decreased yield of *TF*-AgNPs owing to the saturation of the reaction (due to a decreasing number of available Ag ions) [45,46]. 

To further strengthen the results obtained using UV-Vis spectroscopy, the synthesized *TF*-AgNPs were subsequently analyzed using dynamic light scattering studies (DLS) which subsequently can provide a complete inference regarding the Z-average size, also known as the hydrodynamic size (size of the nanoparticle along with the water around the synthesized *TF*-AgNPs), and surface zeta potential (overall charge on the surface of the synthesized AgNPs which provides further information regarding its long-term stability) of the fungal-mediated synthesized *TF*-AgNPs.

Figure 2 shows the dynamic light scattering studies (hydrodynamic diameter and surface zeta potential) of the green-synthesized *TF*-AgNPs. In Figure 2a, it can be seen that fungus-mediated *TF*-AgNPs have an overall diameter of 93.78 nm and carry a surface zeta potential of −25.1 mV. When all particles in the suspension tend to have a large negative zeta potential, they ideally repel each other and have negligible chances of assembling to form aggregates. The large value obtained here suggests that the NPs are well dispersed in solution—this further increases the stability of the biosynthesized formulation of NPs [47]. Furthermore, the synthesized *TF*-AgNPs exhibited a 0.361 poly disparity index (PDI) suggesting that the *TF*-AgNPs synthesized were not aggregated. Similar results were also reported for fungal-biomass-mediated AgNPs [37,48,49,50]. 

Although the DLS studies provided a confirmation regarding the size of the synthesized nanoparticles, they offered a size range with a layer of surface water as the samples are usually dispersed in an aqueous sample. The additional layer of water present on the surface of the nanoparticles signifies a moderate increase in average hydrodynamic diameter of the synthesized nanoparticles; many studies have stressed the importance of the hydrodynamic diameter [51]. Hence, SEM analyses of the dried samples were carried out which not only affected the size of the nanoparticles but also the morphology of the nanoparticles synthesized. Figure 3 shows the appearance of the synthesized *TF*-AgNPs obtained through SEM and TEM. The obtained nanoparticles were spherical in shape and single silver nanoparticles were seen in the electron microscopy images suggesting the absence of agglomeration during the extracellular synthesis of AgNPs. The size range was 5–30 nm with maximum sizes around 20 nm. In a biosynthesis system, the concentration of the reducing agent and presence of stabilizer is crucial for control of the growth of the NPs [52]—the role played here by the fungal biomass.

The information regarding crystalline structure, lattice parameters, grain size, and other characteristics was obtained using XRD; the composition of particles can be obtained by comparing the position and intensity of the peaks with data obtained from the Joint Committee on Powder Diffraction Standards (JCPDS-04-0783) [53]. Figure 4 shows the XRD graph of fungus-mediated synthesized AgNPs. Sharp diffraction peaks were present at 111, 200, 220, 311, 331, and 420 at an angle of 2θ degrees. The emergence of sharp peaks relates to the presence of crystalline nanoparticles. The average size (crystalline) of the synthesized *TF*-AgNPs was 39.76 nm which was calculated using Scherer’s equation with the help of FWHM values of the diffraction peaks. This crystalline grain sizes of these NPs are in agreement with previous reports on bio-based NPs [52,54].

The DLS, Fe-SEM, and XRD analysis provided preliminary information about the size, surface charge, morphology, and phase of the synthesized nanoparticles, which indicated the probable roles of the extracellular and intracellular enzymes present in the endolichenic fungi *T. funiculosus* which acted as reducing and stabilizing agents. Hence, the FTIR analysis was performed to observe the characteristic transmittance bands associated with green-synthesized *TF*-AgNPs and the biomass used for the synthesis. Figure 5 shows the FTIR spectrum of the synthesized *TF*-AgNPs where the presence of specific IR bands was observed. The presence of transmittance bands at 3139, 2317, 1673, 1535, and 1074 cm^−1^ in the synthesized *TF*-AgNPs confirmed the presence of phenolic groups, OH- groups, and C=O groups that may be the functional groups associated with nicotinamide adenine dinucleotide (NADH) and NADH-dependent nitrate reductase, the extracellular enzymes present in most eukaryotic microorganisms. The presence of transmittance bands at 2928, 2329, 1743, and 1062 cm^−1^ in the fungal biomass further confirmed the synthesis of nanoparticles, which provided an insight into the probable role of various intracellular and extracellular enzymes associated with the fungus *Talaromyces funiculosis*—this might act as a capping agent during the synthesis of AgNPs, along with reduction of the metal ion, and contribute to its stability. The values are in agreement with the previously published reports [20,55,56,57] which confirm the synthesis of *TF*-AgNPs in association with fungal biomass. 

### 2.2. Antimicrobial Activity

The current work was anticipated to decipher the antimicrobial effect of green-synthesized AgNPs against diverse pathogenic organisms. The tested *T. funiculosus* functionalized AgNPs exhibited potential antimicrobial activity against both foodborne and human pathogenic bacteria in the preliminary screening using agar well-diffusion methods where the zones of inhibition (ZIs) were measured and *TF*-AgNPs were found to be the most promising antimicrobial agent (Appendix A). Based on the screening results (ZIs), the AgNPs were further tested against all pathogenic organisms in different concentrations using a micro-broth dilution method to determine the minimum inhibitory concentrations (MICs) against individual test pathogens (Figure 6 and Figure 7). 

The *TF*-AgNPs showed substantial effects against all tested microorganisms which were reflected in their MIC (IC_50_) results (Appendix A). The potential antimicrobial activity against tested foodborne pathogens are reflected as MIC values from *S. aureus* (43.94 ± 0.29 μg/mL), *S. faecalis* (52.54 ± 0.16 μg/mL), *L. innocua* (68.94 ± 0.30 μg/mL), and *M. luteus* (45.51 ± 0.21 μg/mL). From the IC_50_ values, it was revealed that the antimicrobial potentials against tested pathogens decreased as follows: *S. aureus > M. luteus > S. faecalis > L. innocua* (Figure 6). Likewise, the antimicrobial activity against human pathogens is represented as MIC values such as for *P. aeruginosa* (21.68 ± 0.72 μg/mL)*, E. coli* (43.94 ± 0.29 μg/mL)*, V. cholerae* (63.69 ± 0.25 μg/mL), and *B. subtilis* (33.60 ± 0.61 μg/mL). The MIC results of *TF*-AgNPs against the human pathogens tested revealed the potentiality of activity as *P. aeruginosa > B. subtulis > E. coli > V. cholerae*. Silver inherently provided antibacterial properties [58]. However, the concentration of Ag used for the synthesis of NPs here was extremely low. Hence, the antibacterial effect of these *TF*-AgNPs was due to a synergistic effect of Ag in combination with fungal metabolites—as observed in previous studies [59,60].

### 2.3. Antioxidant Activity

Oxidative stress due to endogenous factors such as reactive oxygen species (ROS) generation and exogenous factors such as pollution, smoking, pesticides, etc., disrupt the natural balance between natural antioxidants and free radicals produced in the human body. As a result, it is necessary to consume foods enriched with antioxidants—such as from edible fungi, vegetables, and cereals [61]. Figure 8 shows the various antioxidant activity profiles screened from synthesized *TF*-AgNPs. 

The DPPH scavenging assay (Figure 8a) demonstrated that increased concentration of the synthesized *TF*-AgNPs exerted DPPH scavenging at a gradual rate in comparison to the standard ascorbic acid. Another very prominent and toxic reducing agent is nitric oxide whose presence in the cells provides an insight of highly dangerous levels of ROS which can be related to any infection or disease in the body. Figure 8b shows the nitric oxide reducing ability of green-synthesized *TF*-AgNPs. When compared with standard ascorbic acid, the green-synthesized AgNPs showed gradually increased nitric oxide scavenging activity in a dose-dependent manner. This effect of fungus-mediated AgNPs is in agreement with previously published reports of antioxidant activity of green-synthesized Ag nanoparticles [62,63,64]

### 2.4. Hemo-Compatibility Assay

Hemo-compatibility is another important activity for any pharmaceutical substance. Hence, the hemolysis profile of the green-synthesized *TF*-AgNPs was examined using UV-visible spectroscopy (Figure 9). The fungus-mediated synthesized AgNPs exhibited hemolysis < 5% which, according to ISO/TR 7406, provides a safe value for biological molecules [65]. The green-synthesized *TF*-AgNPs can be considered to be reasonably safe for use. The % lyses of positive and solvent controls are also shown in Figure 9.

These results are in accordance with previous reports. A labdane diterpene, copalic acid from *Copaifera* sp., showed high hemolysis (38.4% at 100 Mm); other labdane diterpenes such as 3-hydroxy-copalic and 3-acetoxy-copalic from the same plant were non-hemolytic [66]. The same group of compounds from the seeds of *Alpinia nigra* showed 0.33% hemolysis (conc. 0.44 mg/mL) [67]. However, at higher concentrations most fungi including *Aspergillus* and *Penicillium* show hemolytic potential—hemolysins mostly belonging to a class of proteins from the aegerolysin family [68,69,70]

### 2.5. Cytotoxic Activity Study

The toxicity of NPs, especially AgNPs, is not unheard of—from human health to the environment, higher concentrations of nanoparticles are quite toxic [71]. Figure 10 shows the cytotoxicity efficacy of the synthesized *TF*-AgNPs against MDA-MB-231 breast cancer cells. Ascertaining the cellular cytotoxicity is one of the important parameters for investigating any new formulation or synthesized metallic nanoparticles. Hence, in the present study, the biocompatibility and cytotoxicity level of our fungus-mediated synthesized AgNPs were studied using MDA-MB-231 breast cancer cells. The formation of insoluble formazan from the MTT dye (3-(4, 5-dimethylthiazol-2-yl)-2, 5-diphenyltetrazolium bromide) through the action of the mitochondrial enzyme NAD (P) H-dependent cellular oxidoreductase directly correlates to the level of cellular metabolism. 

As depicted in Figure 10a, the synthesized *TF*-AgNPs exhibited cytotoxicity in a dose-dependent manner against MDA-MB-231 breast carcinoma cells with a significant decrease in viable cells with increase in AgNPs concentration. The obtained results are in congruence with previously published reports—biosynthesized AgNPs have proved to be cytotoxic to different cancerous cell lines such as HepG2, A549, and MCF-7; however, they are almost non-toxic to non-cancerous or normal Vero cell lines [30,72,73]. To further validate our findings from the MTT assay, a chromatin condensation assay was carried out using Hoechst 33342 staining. Figure 10b(ii) shows the formation of granular condensed nuclei of the TF-AgNPs-treated MDA-MB-231 breast carcinoma cells. To further investigate the cellular viability and migration potential of the synthesized nanoparticles, a scratch wound healing assay was performed. From Figure 11, we can clearly observe that after 12 and 24 hours of treatment with the synthesized AgNPs, the migratory capacity of the MDA-MB-231 breast carcinoma cells decreased subsequently as the scratch formed within the Petri plate had a significant distance. Similar results were reported earlier by Nayak’s group upon treatment with plant-mediated synthesized AgNPs [70,74].

### 2.6. In Vitro ROS Activity

In cancer therapy, the intracellular production of ROS is emerging as a new alternative to traditional chemotherapy. During cancer prognosis and advancement, the reactive oxygen species inside the cancerous cells targets nearby normal cells and attack them, thereby changing their dynamics and amalgamating the nearby healthy cells into their group of cancerous tissues. A wide-ranging assortment of free radicals comprising hydrogen peroxide (H_2_O_2_), nitric oxide (NO), superoxide anion (O_2_^−^), peroxynitrite (ONOO-), hydrochlorous acid (HOCl), and hydroxyl radicals (OH-) are produced intracellularly. The DCFH-DA dye provides the inference of the overall conglomeration of intracellular ROS produced in the cells. When the dyes interact with the ROS radicals accumulated inside the cancerous cells upon treatment with the AgNPs, they become sensitive and produce various levels of fluorescence intensity which is ultimately recorded using a flow cytometer and fluorescence microscopy. Figure 12 shows the fluorescence peaks of the control and AgNPs treated through flow cytometry and fluorescence microscopy. 

It can be clearly observed that the fluorescence intensity of the nanoparticle-treated cells is higher than the control group (Figure 12). The increased generation of ROS after AgNP treatment provides an insight into their possible role in overall damage of the cellular machinery such as DNA, the nucleus, and the mitochondrial membrane of the cancerous cells. This apoptotic potential of cancerous cells due to increased ROS generation after treatment with AgNPs has also been highlighted in previous reports [75,76].

### 2.7. Predation of Mesocyclops aspericornis against A. stephensi and Aedes aegypti

It is interesting to note that the potential predation of mosquito larvae by copepods, *M. aspericornis,* was evidenced during the treatment with silver nanoparticles (*TF*-AgNPs). When there were silver nanoparticles, the mosquito movement was slowed and at that time, the predation rate/percentage was high. Moreover, the physiological status of the larvae was disturbed. Under laboratory conditions, the predatory efficiency per copepod per day was 3.47, 2.04, 0.76, and 0.23 larvae for *A. stephensi* and 5.75, 4.73, 2.3, and 0.90 larvae for *A. aegypti* (I, II, III, and IV, respectively) (Table 1). In an AgNP contaminated environment, the predatory efficiency (predatory potential/No. of predation/ability of predation) of one copepod per day was 4.92, 2.96, 1.52, and 0.81 larval instars of *Anopheles* species and 7.16, 5.97, 3.30, and 1.99 larvae for the dengue vector (first, second, third, and fourth instars, respectively) (Table 2). Murugan et al. [77] reported a considerable predation of *M. aspericornis* towards the control of young stages of *A. aegypti.* The application of a lower dose (i.e., 1 ppm) of AuNPs synthesized with lemongrass may help to control the malaria and dengue vectors using predatory copepods [78].

## 3. Materials and Methods

### 3.1. Materials and Reagents

Chemicals such as 3-(4, 5-dimethylthiazol-2-yl)-2,5 di-phenyl-tetrazolium bromide (MTT), modified Eagle’s medium (MEM), fetal bovine serum (FBS), antibiotic solution (penicillin–streptomycin), dichloro-dihydro-fluorescein diacetate (DCHF-DA), Hoechst 33342 stain, and cisplatin were purchased from Sigma-Aldrich (Mumbai, India). The other essential media and analytical-grade reagents such as Mueller–Hinton media and silver nitrate (AgNO_3_) were obtained from Hi-Media (Mumbai, India). Milli-Q water was employed while preparing any chemical solution during the experiments. 

### 3.2. Cell-Free Biomass Extract Preparation

Pure cultures of *T. funiculosus* (laboratory isolates) were grown on Potato Dextrose Broth (PDB) Erlenmeyer flasks. The inoculated flasks were kept in a BOD shaking incubator (14 days, 29 ± 1 °C) with intermittent shaking (150 rpm). After growth was completed, the cultures were passed through a sterile cheesecloth to eradicate the mycelial mats and mycelia-free liquids were kept in separate sterile flasks or tubes [55]. The tubes were composed of a suitable material to carry out vigorous centrifugation to remove cell debris present inside the liquid broth to make it an optimum cell-free extract. The *T. funiculosus* cell-free extract was kept in 4 °C for further use.

### 3.3. Synthesis of AgNPs

The *T. funiculosus* cell-free extract was added to 1 mM silver nitrate solution (1:1) along with the control and the reaction mixtures were incubated in a rotary shaker at 200 rpm and 28 °C in dark conditions in triplicates. After the incubation was completed, the color changed to dark brown and the whole mixture solution was centrifuged (10,000 rpm, 15 min). The supernatant was then decanted, and the pellet was taken for lyophilization. The dried lyophilized AgNP powder was kept at 4 °C for further study [41].

### 3.4. Characterization Studies

For characterization of the synthesized AgNPs, the color change solution was subjected to scanning in a UV-Vis spectrophotometer (Lambda 35R PerkinElmer, Waltham, MA, USA) in the range of 200–800 nm. The dynamic light scattering properties of the AgNPs were measured using a Zetasizer (ZS 90, Malvern Instruments Ltd., Malvern, UK). Briefly, the DLS studies provide the hydrodynamic diameter, surface zeta potential, and poly disparity index (PDI) where the samples were dispersed in water and by using a 1 mL syringe they were poured into the sample holder. The samples were scanned using the Malvern ZS nano software. The surface morphology and the size of the AgNPs were studied using SEM (Nova Nano SEM 450/FEI, Lincoln, NE, USA) and TEM (Technai™ F30 G2 STWIN, FEI, Lincoln, NE, USA). To examine the crystalline structure and pattern of the synthesized AgNPs, an X-ray diffractometer (PANalytical X’Pert, Almelo, The Netherlands) equipped with a Ni filter and a Cu Kα (l = 1.54056 Å) radiation source was used and the scanning range was between 20 and 80° at 0.05°/s. To analyze the molecular and surface modification of the synthesized AgNPs, ATR-FTIR spectroscopy (Bruker ALPHA spectrophotometer, Ettlinger, Germany) was used. Briefly, the sample (single drop) was kept on the sample holder and scanned between 4000 and 500 cm^−1^ at a resolution of 4 cm^−1^. The obtained result was then analyzed using OPUS software (Opus 7.0, Bruker, Billerica, MA, USA).

### 3.5. Antimicrobial Activity

The antibacterial test of the AgNPs synthesized using a green route was performed against the foodborne pathogens *Staphylococcus aureus*, *Streptococcus faecalis*, *Listeria innocua,* and *Micrococcus luteus* and some medically important nosocomial bacterial strains: *Pseudomonas aeruginosa, Escherichia coli, Vibrio cholerae*, and *Bacillus subtilis*. The bacterial strains were procured from MTCC, India. Briefly, isolated colonies of test organisms were inoculated in a flask containing 20 mL of sterilized MHB (Mueller–Hinton broth) and were incubated (24 h, 37 °C); *B. subtilis* was incubated at 30 °C. Post incubation, the turbid test culture was spread onto MHA plates in triplicates using a disposal spreader and wells were bored in each plate using a sterilized cork borer (5 mm). Finally, 100 µL of aqueous suspension of AgNPs (concentration of 100 µg/mL) were disposed into each well and incubated (24 h, 37 °C).

Following the agar well-diffusion methods, the micro-broth dilution (MBD) method was employed to calculate the minimum inhibitory concentrations (MICs) of AgNPs regarding antimicrobial activities [43]. The MBD method for microbial growth or inhibition was carried out in 96-well plates and measured with a Microplate Reader (Biorad, Hercules, CA, USA) at 600 nm. The MIC was evaluated and expressed as IC_50_ by employing IC_50_/IC_90_ Laboratory Excel Calculation Tools. 

### 3.6. Antioxidant Activity

#### 3.6.1. DPPH Scavenging Assay

The antioxidant activity of the green-synthesized AgNPs was estimated using the DPPH scavenging assay as previously described [43]. Briefly, 1 mL of DPPH solution (4 mg in 100 mL methanol) was added to different concentrations viz 10, 20, 30, 40, 50, 60, 70, 80, 90, and 100 µg/mL of *TF*-AgNPs. The DPPH-radical-added reaction mixture was incubated in dark conditions for 30 min and the absorbance was measured at 517 nm. The ascorbic acid was used as the positive control. The scavenging free-radical activity was determined using the formula:$$Scavenging\;free\;radical\;activity\;\%=\frac{Ac-As}{Ac}\;\times \;100$$
where **Ac** is the absorbance of the control and **As** is the absorbance of the sample/standard.

#### 3.6.2. Nitric Oxide (NO) Scavenging Activity

The [NO] scavenging potential of *TF*-AgNPs was evaluated by following the method described previously [79] employing the Griess Illosvoy reaction. Briefly, 1 mL of the *TF*-AgNPs in various concentrations (viz., 10–100 µg/mL) was mixed with sodium nitroprusside (1 mL) and incubated (25 °C, 150 min). After incubation, 1 mL of the solution was added with Griess reagent (1 mL) and was further incubated (30 min) and the absorbance was measured at 546 nm.

### 3.7. Hemolytic Assay

The hemolysis activity of the green-synthesized *TF*-AgNPs were analyzed using a standard protocol [74]. Briefly, goat blood was collected from a local butchery and was diluted with normal saline buffer. Then, 10 mL of *TF*-AgNPs (1 mg/mL) was added to 10 mL normal saline and after adding 0.2 mL of diluted blood, and it was incubated at 30 °C for 1 h. After post-incubation, the reaction mixture was centrifuged (300 rpm, 5 min) and the absorbance of the supernatant was measured at 500 nm. The diluted blood (0.2 mL) was added to 10 mL of 0.1% sodium carbonate as a positive control and 10 mL of normal saline solution as a negative control. The % hemolysis rate was determined using the following formula: $$HEMOLYSIS\;\left(\%\right)=\frac{ODt-ODnc}{ODpc-ODnc}$$

### 3.8. Cytotoxic Activity

#### 3.8.1. MTT Assay

The effect of green-synthesized AgNPs against MBA-MB 231 breast carcinoma cells was determined using a colorimetric MTT assay using an accepted protocol [80,81]. Briefly, MBA-MB 231 cells (breast cancer cell line) were procured from a cell line culture collection (NCCS, Pune, India). The procured cells were plated in 96-well plates using the most commonly accepted medium (MEM, 10% FBS, 1% penicillin–streptomycin antibiotic solution) under standard conditions. After achieving 70% confluence, different concentrations of *TF*-AgNPs (viz., 10–200 µg/mL) were added to cells and incubated for 24 h. To assess the cell viability, 100 µL of MTT (5 mg/mL) was added to the cells of each well and were subsequently incubated for 4 h. After post-incubation, the MTT solution was removed and to each well DMSO (100 μL) was added and kept in the dark for 15 min. After the stipulated time, the OD of the formazan product was measured at 595 nm. 

#### 3.8.2. Chromatin Condensation Assay with Hoechst 33342 Staining

The DNA chromatin condensation assay was performed with Hoechst 33342 staining using a standard protocol [77]. In a 60 mm Petri plate, the MDA-MB-231 breast carcinoma cells were plated. After achieving 70% of cell confluence, *TF*-AgNPs were treated with the cells and after post-incubation for 24 h, the cells were stained with 1 mg/mL Hoechst 33342 stain and were further incubated (10 min, 37 °C). The cells were captured using fluorescence microscopy under a UV filter using Epifluorescent Microscopy (Olympus IX71, Olympus, Tokyo, Japan)

#### 3.8.3. Scratch Wound Healing Assay

For a scratch wound healing/migration assay, the MDA-MB-231 breast carcinoma cells were transferred to a 6-well Petri plate. At 90% confluency, a scratch was made using a sterile micro tip. After scratching, the medium was replaced with a medium containing *TF*-AgNPs and pictures were captured at different incubation time periods (0, 12, and 24 h).

#### 3.8.4. In Vitro ROS Activity

The ROS activity of the *TF*-AgNPs was evaluated using a standard dichloro-dihydro-fluorescein diacetate (DCFH-DA) assay [82]. The MDA-MB-231 cells were treated with *TF*-AgNPs and then subsequently treated with DCFH-DA dye. The complete reaction mixture was incubated for 4 h using flow cytometry (FL-1 filter) and fluorescence microscopy and the intracellular ROS activity was monitored. 

### 3.9. Predation of M. aspericornis against Malaria and Dengue Mosquitoes 

Mosquito-predating copepods were experimented against mosquito larvae of malaria and dengue vectors in contaminated and non-contaminated nanoparticle environments. There were 100 mosquito larvae (stages I to IV) for 10 adult copepods. Experiments were conducted in 500 mL of dechlorinated water in a glass beaker. Mosquito larvae were replaced with new ones on a daily basis throughout the experimentation. There were 5 replications for each experiment and 250 mL of dechlorinated water without copepods was used as a negative control. Experimental beakers were evaluated for 5 days and prey consumption was also noted. The predatory efficiency was evaluated using the standard formula:Predatory efficiency = number of consumed mosquitoes/number of predators/total number of mosquitoes × 100

### 3.10. Predation of M. aspericornis against Malaria and Dengue Mosquitoes Post Treatment with AgNPs

In this experiment, we evaluated larval control by adult copepods with silver nanoparticles. In total, 100 larvae were used for 10 copepods in 500 mL water with *TF*-AgNPs (i.e., for both species, 1/3 of the LC_50_ was calculated against the Ist larval stage). There were five replications and new larvae were introduced daily. 

### 3.11. Statistical Analysis

All experiments were carried out in triplicates, and the data were presented as means ± standard deviation using IBM Corporation (Armonk, NY, USA), 2012 Statistics, and all graphs were illustrated using Origin 6.0. 

## 4. Conclusions

Silver nanoparticles are considered as the magic bullet in the 21st century owing to their outstanding exposure in every field of technology and medical and healthcare industries. The use of organisms such as fungi for the synthesis of AgNPs provides additional functional groups, thereby increasing the novelty of the nanoparticles. Since breast carcinoma is a major cause of concern for women worldwide, an efficient and cost-effective alternative is of utmost importance. The *T. funiculosus*-functionalized AgNPs synthesized through the greener route without any additional reducing and capping agents exhibited dose-dependent cytotoxicity against MDA-MB-231 breast carcinoma cell lines. The in vitro ROS production and fluorescence microscopy analysis through Hoechst 33342 staining showed that the biogenic-synthesized AgNPs induced chromatin condensation after their treatment in breast carcinoma cells. The intracellular ROS production provides a cascade of events that ultimately targets the DNA machinery and the mitochondrial activity of the cancerous cell, thereby causing the overall disintegration of DNA replication, protein synthesis, and ATP production inside the cancerous cell. Furthermore, the scratch migration assay also supported the inhibition of the overall migratory potentials of the MDA-MB-231 breast carcinoma cells by the activity of the AgNPs. However, proper in vivo experiments are necessary to evaluate the anti-cancerous activity of the synthesized AgNPs for proper validation and clinical trials. The development of novel tools for mosquito control is of prime importance for an effective and reliable integrated vector management strategy design. Biofabricated nanoparticles from *T. funiculosus* are the most effective biomolecules for the vector control program. Furthermore, the use of a low dose of fungal bio-insecticides together with biopredation is a promising green pathway for effective mosquito management processes in practice. 

## Figures and Tables

**Figure 1 ijms-23-10626-f001:**
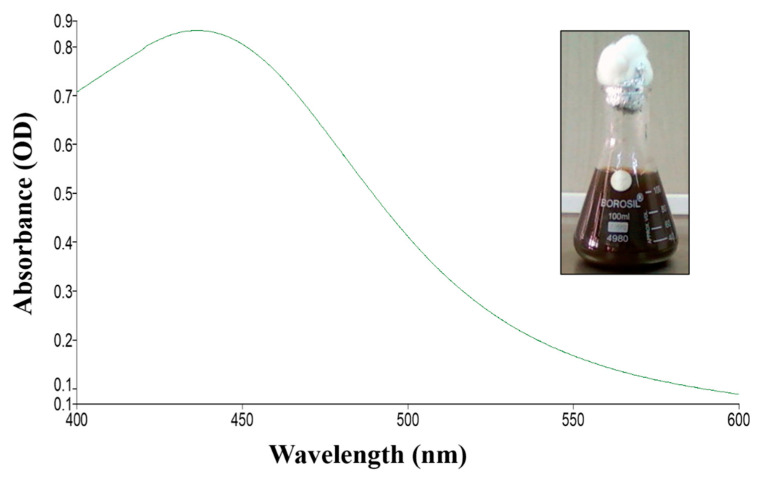
UV-Visible spectrophotometric analysis of AgNPs synthesized using *Talaromyces funiculosus*.

**Figure 2 ijms-23-10626-f002:**
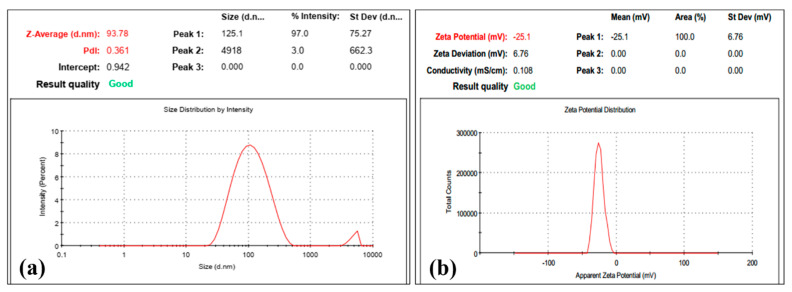
DLS of AgNPs synthesized using *T. funiculosus* extracts. (**a**) Distribution of the average size. (**b**) Surface charge.

**Figure 3 ijms-23-10626-f003:**
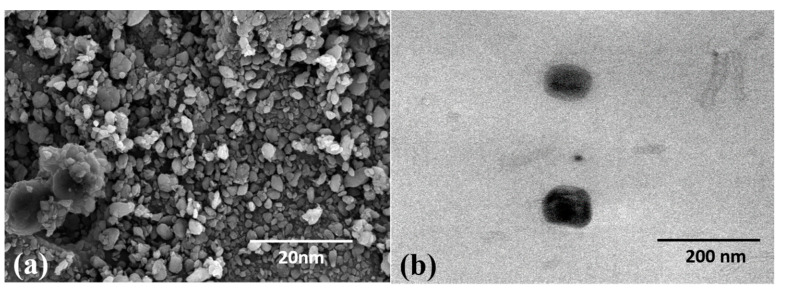
(**a**) SEM micrograph and (**b**) TEM micrograph of AgNPs synthesized using *T. funiculosus* culture filtrate.

**Figure 4 ijms-23-10626-f004:**
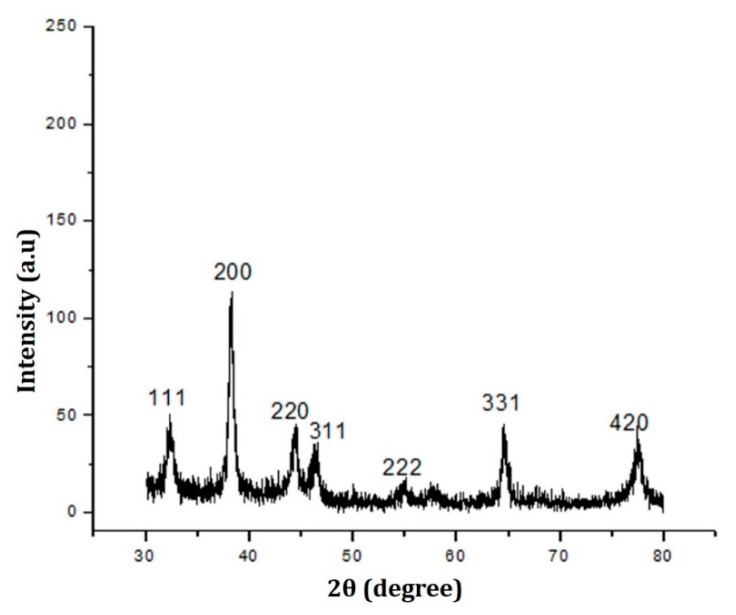
XRD diffractogram of *Talaromyces funiculosus*–AgNPs.

**Figure 5 ijms-23-10626-f005:**
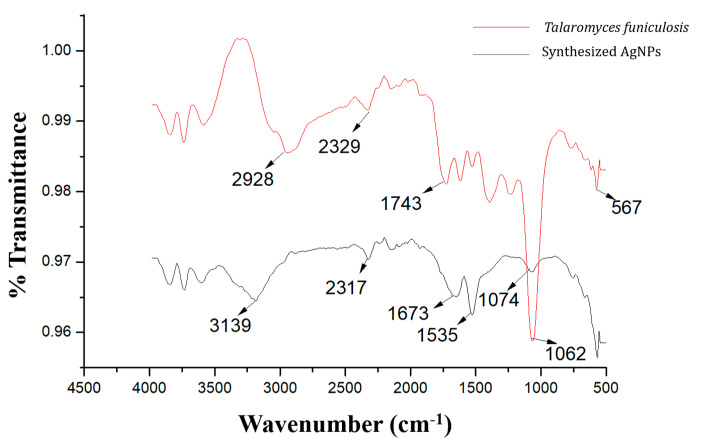
ATR-FTIR spectrum of *T. funiculosus* extract and *T. funiculosus*–AgNPs.

**Figure 6 ijms-23-10626-f006:**
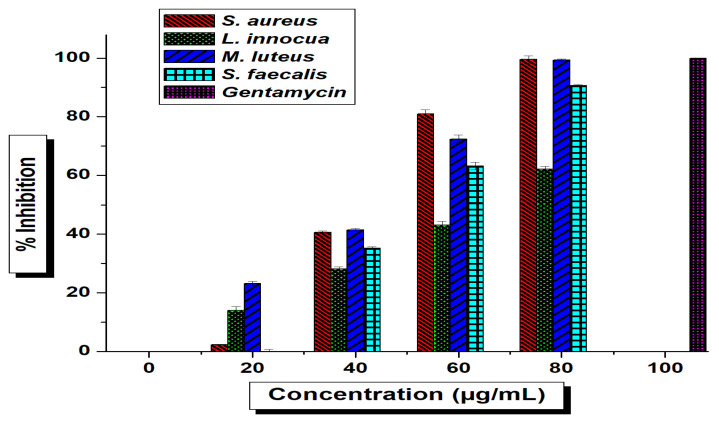
Determination of MICs of *T. funiculosus*–AgNPs against foodborne pathogens using microbroth dilution; standard deviation is shown. Significant difference (*p* ≤ 0.05).

**Figure 7 ijms-23-10626-f007:**
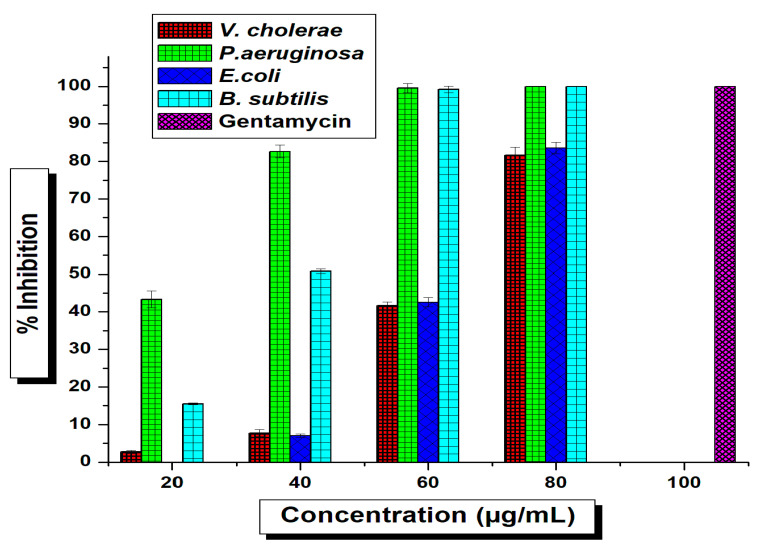
Determination of MICs of *T. funiculosus*–AgNPs against human pathogenic bacteria using microbroth dilution; the standard deviation shown with error bars indicates significant differences (*p* ≤ 0.05).

**Figure 8 ijms-23-10626-f008:**
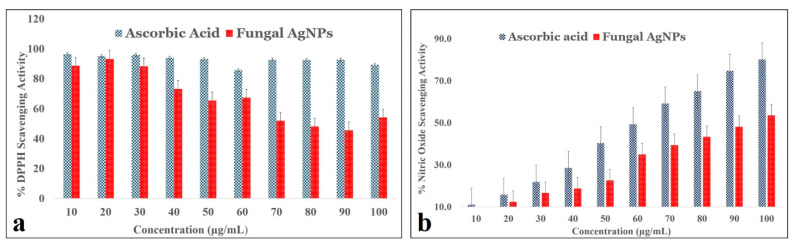
Antioxidant activity of *T. funiculosus*–AgNPs in terms of radical scavenging activity: (**a**) DPPH scavenging activity, (**b**) nitric oxide scavenging activity. Error bars represent standard deviation of the mean. Significant difference (*p* ≤ 0.05).

**Figure 9 ijms-23-10626-f009:**
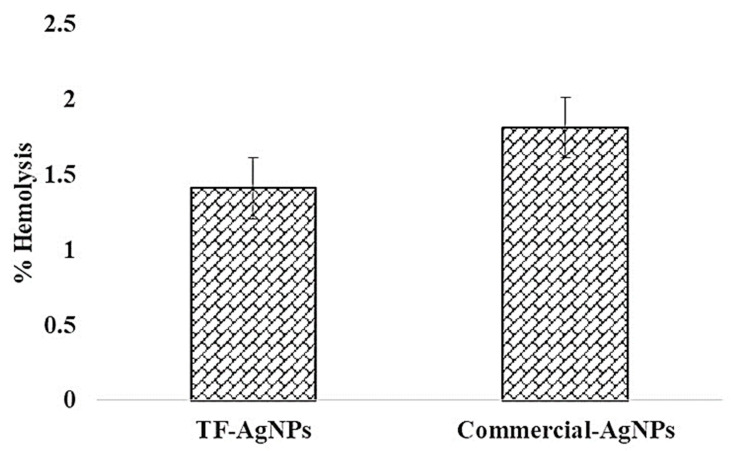
Hemolysis activity of the synthesized *T. funiculosus*–AgNPs and commercial AgNPs.

**Figure 10 ijms-23-10626-f010:**
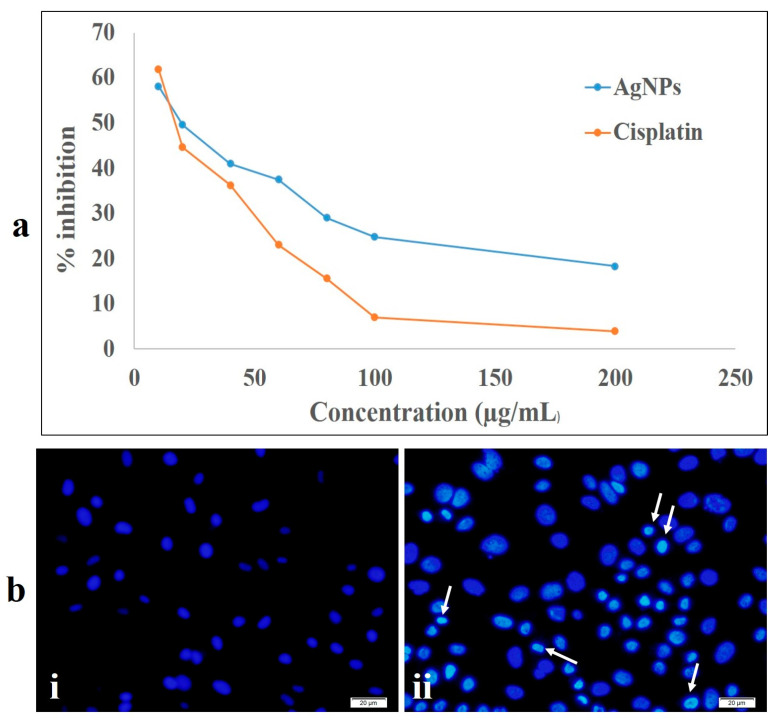
Anticancer activity of the synthesized *T. funiculosus*–AgNPs against MDA-MB-231 breast cancer cell lines. (**a**) MTT assay; (**b**) Hoechst 33342 staining: (i) untreated cells; (ii) cells treated with *TF*-AgNPs (white arrows indicate the condensed nuclear material).

**Figure 11 ijms-23-10626-f011:**
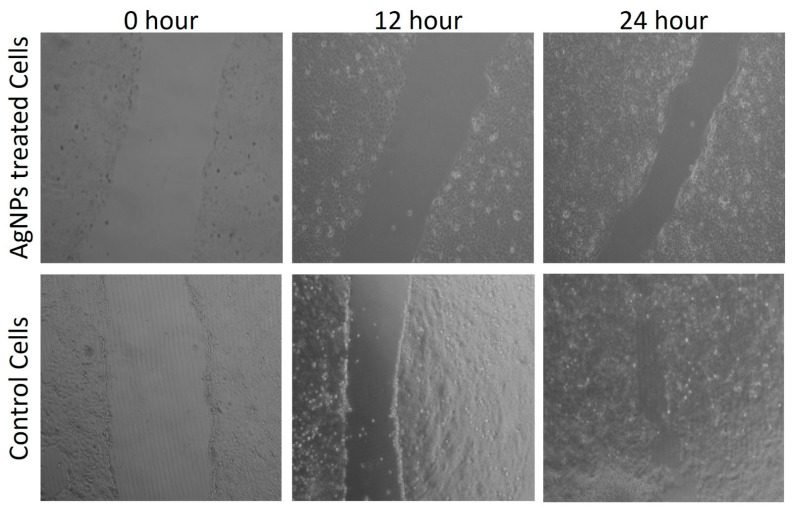
Scratch wound healing assay of the green-synthesized *T. funiculosus*–AgNPs against MDA-MB-231 breast cancer cells and control cells.

**Figure 12 ijms-23-10626-f012:**
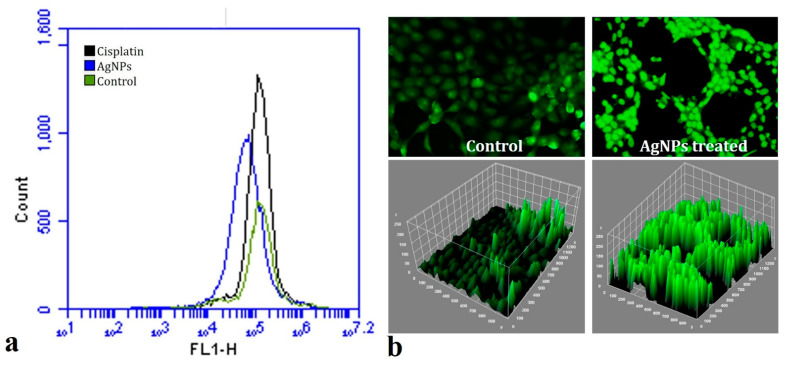
(**a**) Flow cytometric analysis of ROS production (in vitro) in MDA-MB-231 cells after AgNP treatment; (**b**) fluorescence microscopy images of the MDA-MB-231 cells treated with AgNPs analyzed using **ImageJ** software.

**Table 1 ijms-23-10626-t001:** Predation efficiency of the copepod *Mesocyclops aspericornis* against larvae of *Anopheles stephensi* and *Aedes aegypti*.

Mosquito Species	Targeted Instar	Number of Consumed Prey						Total Predation (n)	Consumed Prey per Copepod per Day
		Control	Day 1	Day 2	Day 3	Day 4	Day 5		
** *Anopheles stephensi* **	**Larva I**	0	36.6 ± 2.6	37.2 ± 1.0	35.6 ± 1.5	33.6 ± 3.5	30.5 ± 1.2	173.5	3.47
	Larva II	0	22.8 ± 0.5	20.8 ± 0.8	19.2 ± 0.8	18.9 ± 2.1	20.4 ± 1.6	102.1	2.04
	Larva III	0	8.1 ± 1.9	7.6 ± 1.1	7.1 ± 0.4	7.5 ± 1.5	7.8 ± 1.0	38.1	0.76
	Larva IV	0	2.3 ± 2.1	2.2 ± 1.0	2.2 ± 1.7	2.6 ± 1.6	2.5 ± 1.3	11.8	0.23
** *Aedes aegypti* **	Larva I	0	60.3 ± 1.8	58.1 ± 2.8	55.4 ± 1.0	57.9 ± 0.7	56.1 ± 2.4	287.8	5.75
	Larva II	0	49.2 ± 1.4	47.6 ± 2.4	45.2 ± 0.9	47.8 ± 2.4	47.1 ± 1.8	236.9	4.73
	Larva III	0	26.1 ± 1.8	24.3 ± 0.5	22.4 ± 2.0	21.3 ± 3.1	20.9 ± 1.0	115	2.3
	Larva IV	0	9.5 ± 0.6	8.9 ± 1.8	8.6 ± 2.6	8.9 ± 2.4	9.2 ± 1.1	45.1	0.90

Predation efficiency results with means ± SD of five replicates (10 copepods vs. 100 mosquitoes per replicate). The control was clean water without copepods (*p* < 0.05).

**Table 2 ijms-23-10626-t002:** Predation efficiency of the copepods against larvae of *Anopheles stephensi* and *Aedes aegypti* after treatment with TF-AgNPs.

Mosquito Species	Targeted Instar	Number of Consumed Prey						Total Predation (n)	Consumed Prey per Copepod per Day
		Control	Day 1	Day 2	Day 3	Day 4	Day 5		
** *Anopheles stephensi* **	**Larva I**	0	49.7 ± 1.5	50.4 ± 1.8	49.5 ± 2.5	47.9 ± 1.5	48.6 ± 3.1	246.1	4.92
	Larva II	0	30.2 ± 0.9	27.6 ± 1.6	29.2 ± 3.1	31.4 ± 2.7	30.7 ± 2.1	149.1	2.98
	Larva III	0	14.6 ± 1.1	15.3 ± 1.9	14.7 ± 2.7	16.1 ± 1.6	15.5 ± 1.6	76.2	1.52
	Larva IV	0	8.8 ± 1.1	9.1 ± 0.4	7.4 ± 1.3	7.4 ± 1.8	8.2 ± 1.9	40.9	0.81
** *Aedes aegypti* **	Larva I	0	72.3 ± 1.5	70.4 ± 0.7	73.1 ± 0.9	70.5 ± 0.5	71.9 ± 2.1	358.2	7.16
	Larva II	0	59.1 ± 1.9	62.7 ± 1.8	59.0 ± 1.4	57.9 ± 1.4	60.1 ± 1.0	298.8	5.97
	Larva III	0	33.4 ± 2.4	35.1 ± 0.4	30.9 ± 1.4	33.4 ± 2.1	32.6 ± 1.2	165.4	3.30
	Larva IV	0	20.1 ± 0.9	18.9 ± 2.7	19.5 ± 0.8	20.1 ± 2.6	20.9 ± 2.3	99.5	1.99

Predation rates are means ± SD of five replicates (10 copepods vs. 100 mosquitoes per replicate). The control was clean water without copepods (*p* < 0.05).

## Data Availability

Additional data to those presented here are available from the first author upon reasonable request.

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
