# Peer review of "Green Synthesis of Endolichenic Fungi Functionalized Silver Nanoparticles: The Role in Antimicrobial, Anti-Cancer, and Mosquitocidal Activities"

_ijms, 2022, doi:10.3390/ijms231810626_

Round 1
Reviewer 1 Report
Review comments of ijms-1851154 manuscript
Current manuscript was entitled “Green synthesis of endolichenic fungi functionalized silver nanoparticles: Role in antimicrobials, anti-cancer and mosquitocidal activities”. Authors have demonstrated their results with potential experimental evidence, so I would recommend this manuscript for publication after fulfilling the following minor comments.
1. Authors have to change the references format, as according to the journal guidelines.
2. In figure 3b, authors have to change the figure caption to TEM micrograph instead of HR-TEM micrograph.
3. In figure 4, authors have to change to x-axis is 2θ (degree) and y- axis is Intensity (a.u.) of the AgNPs XRD pattern and compared with the JCPDS number.
4. In figure 5, authors have to identify the respective functional groups of T. funiculosus AgNPs in the figure.
5. In the text, authors have written as Fe-SEM instead of SEM.
6. In figure 6&7, the supplementary table could not find. Authors have to demonstrate their antibacterial activity along with optical images.
7. In figure 10a, authors must distinguish the x and y axis (MTT assay plotting was unclear). And in figure 10b, what was the difference between two fluorescence images?
8. In figure 11, authors have to include the wound healing plot at different time points.
9. In 3.6. In vitro ROS activity, the paragraph entire paragraph has to be rewrite with the adequate explanation. In figure 12a, why authors treated the cancer cell with drug (Cisplatin) to the flow cytometry analysis?

Author Response
Reviewer#1
Current manuscript was entitled “Green synthesis of endolichenic fungi functionalized silver nanoparticles: Role in antimicrobials, anti-cancer and mosquitocidal activities”. Authors have demonstrated their results with potential experimental evidence, so I would recommend this manuscript for publication after fulfilling the following minor comments.
Comment#1: Authors have to change the references format, as according to the journal guidelines. Response: The references have been formatted using Mendeley software according to the journal’s guidelines.
Comment#2: In figure 3b, authors have to change the figure caption to TEM micrograph instead of HR-TEM micrograph.
Response: We have changed the figure caption of 3b to TEM.
Comment#3: In figure 4, authors have to change to x-axis is 2θ (degree) and y- axis is Intensity (a.u.) of the AgNPs XRD pattern and compared with the JCPDS number.
Response: We have changed the legends of the X and Y axis of figure 4 XRD image. In the result section the data have been compared with the JCPDS number.
Comment#4: In figure 5, authors have to identify the respective functional groups of T. funiculosus AgNPs in the figure.
Response: The respective functional groups and their attributes have been incorporated in the modified manuscript in the FTIR section of results.
Comment#5: In the text, authors have written as Fe-SEM instead of SEM.
Response: The typological errors have been modified accordingly.
Comment#6: In Figures 6&7, the supplementary table could not find. Authors have to demonstrate their antibacterial activity along with optical images.
Response: The Supplementary file of the revised version provides Tables and Figures of the Antimicrobial tests.
Comment#7: In figure 10a, authors must distinguish the x and y axis (MTT assay plotting was unclear). And in figure 10b, what was the difference between two fluorescence images?
Response: The omission of legends for X and Y axis of the MTT graph is highly regrettable. In the revised manuscript we have incorporated all the points raised by the reviewer.
Figure 10b figure shows the difference in chromatin condensation activity in the control of untreated cells and the treated cells. The Hoechst 33342 stain primarily stains the condensed nuclear material as after treatment with nanoparticles the nuclear envelope is destroyed and the stain can easily pass through the damaged nuclear membrane. In control cells, the nuclear membrane is intact since the stain cannot pass through it and bind with the nuclear material.
Comment#8: In figure 11, authors have to include the wound healing plot at different time points.
Response: In Figure 11 we have observed the scratch wound healing assay of the green synthesized T. funiculosus–AgNPs against MDA-MB-231 breast cancer cells at (a) Zero hour incubation; (b) 12 hour incubation; (c) 24 hour of incubation.
Comment#9: In 3.6. In vitro ROS activity, the paragraph entire paragraph has to be rewrite with the adequate explanation. In figure 12a, why authors treated the cancer cell with drug (Cisplatin) to the flow cytometry analysis?
Response: Taking the suggestions of the reviewer, we have modified the 3.6 in vitro ROS activity section with adequate references. Cisplatin is a standard FDA approved anticancer drug, here in this experiment of flow cytometry Cisplatin was used as a standard to observe the difference in internal ROS activity when treated with AgNPs and Cisplatin. Many groups have reported a similar type of results for anticancer drug examination (Zaidieh, T., Smith, J.R., Ball, K.E. et al. ROS as a novel indicator to predict anticancer drug efficacy. BMC Cancer 19, 1224 (2019). https://doi.org/10.1186/s12885-019-6438-y)

Reviewer 2 Report
Mohanta et al present in this manuscript the production of silver NPs through a "green synthesis approach" using a Talaromyces funiculosus based approach. Overall the manuscript is well written with certain parts though especially in the discussion part that need improvement. The experimental part employs a wide series of techniques especially the part about the characterization studies where I believe a holistic approach was followed. There is an obvious lack of details regarding the assays followed in order to describe the biomedical aspect of NPs but the truth is that there is a limit in the number of the assays that could be applied in the context of a project.
Below you could find a list with a revisions that I would like to be addressed by the authors so the manuscript can be accepted:
line 150: 2.1 Materials and Reagents paragraph. some additional details about the reagents used would be useful. e.g. the full name and the product code numbers would be necessary.
line 192: 2.5 Antimicrobial activity paragraph. Please include the MTCC No of each pathogen used. Also, regarding the cultivation conditions for each pathogen, did you follow aerobic anaerobic conditions? you mentioned that you cultivated all pathogens at 37oC, what about Bacillus subtilis did you use 37 or 30oC?
lines 211-220 please provide details (company, product code) for the reagents/kits used
lines 245: please provide code Number of the cell line
line 275: what is the stage/age of the larvae used?
Figures
Figure 8: the are no units mentioned
Figure 9: check font/graphs size. Are the numbers on the y axis correct? e.g. maximus hemolysis 0.018 % is that correct? or it should be 1.8 %. Also what concentration of NPs did you use here?
Figure 10: there are no details on the axis, there no titles, no information about what the numbers represent. Also a legend is necessary.
Figure 12: you need to describe in the legend what cisplatin is and why did you use that. Also this reagents needs to be included and described in the materials and methods sections. You need also to include a non-treated control in figure 12a like what you did in figure 12b.
Additional issues:
line 522: it is known that there is potential predation of mosquito larvae by copepods. the question and the potential contribution of your work is if the presence of silver NPs facilitate this predation.
line 523: efficiency of what?
line 561-563: you need to rewrite this part. line 561, the most effective is too ambitious. there are several ways to eliminate mosquitos. you just showed that the presence of NPs slightly improved the action of copepods. line 562-563 I can't follow that and I don't understand this conclusion from the authors.
Finally a major issue is related to the scratch migration assay. the authors should provide a control and then compare the healing with or without NPs and include the control as extra panels in Figure 11 for 0, 12, and 24 hour. As soon as you include the control and after comparing the images with NPs you can then say what you claim in lines 489-490 about the migratory capacity of the carcinoma cells.
Author Response
Reviewer #2
Mohanta et al present in this manuscript the production of silver NPs through a "green synthesis approach" using a Talaromyces funiculosus based approach. Overall the manuscript is well written with certain parts though especially in the discussion part that need improvement. The experimental part employs a wide series of techniques especially the part about the characterization studies where I believe a holistic approach was followed. There is an obvious lack of details regarding the assays followed in order to describe the biomedical aspect of NPs but the truth is that there is a limit in the number of the assays that could be applied in the context of a project. Below you could find a list with a revisions that I would like to be addressed by the authors so the manuscript can be accepted:
Comment#1: line 150: 2.1 Materials and Reagents paragraph. some additional details about the reagents used would be useful. e.g. the full name and the product code numbers would be necessary.
Response: The full names of the reagents used were added in the materials and methods section.
of the revised version.
Comment#2: line 192: 2.5 Antimicrobial activity paragraph. Please include the MTCC No of each pathogen used. Also, regarding the cultivation conditions for each pathogen, did you follow aerobic anaerobic conditions? you mentioned that you cultivated all pathogens at 37oC, what about Bacillus subtilis did you use 37 or 30oC?
Response: We have added the MTCC No of the bacterial samples used. There was a typographical mistake on the incubation temperature of B. subtilis, for Bacillus subtilis the incubation temperature used was 30 oC.
Comment#3: lines 211-220 please provide details (company, product code) for the reagents/kits used.
Response: The Full form of the reagent used along with their company name has been included in the revised manuscript.
Comment#4: lines 245: please provide code Number of the cell line
Response: The cells were procured from NCCS Pune (National Centre for Cell Science Complex, Savitribai Phule Pune University Campus, Ganeshkhind Road, Pune - 411007, India, Maharashtra). In the NCCS Pune they don’t provide any code number for cell lines, the cells are sent through registered Speed Post, Govt. of India Post upon receipt of Registration form and Payment Demand Draft.
Comment#5: line 275: what is the stage/age of the larvae used?
Response: I to IV stages of mosquito larvae were used and mentioned in the respective section text.
Comment#6: Figure 8: there are no units mentioned
Response: In figure 8 the legends in the X and Y axis have been incorporated.
Comment#7: Figure 9: check font/graphs size. Are the numbers on the y axis correct? e.g. maximus hemolysis 0.018 % is that correct? or it should be 1.8 %. Also what concentration of NPs did you use here?
Response: The graph was plotted again and the % hemolysis value is 1.4% and 1.71 % respectively for TF-AgNPs and Commercially available AgNPs from Sigma Aldrich. The concentration of AgNPs used was 1 mg/mL
Comment#8: Figure 10: there are no details on the axis, there no titles, no information about what the numbers represent. Also a legend is necessary.
Response: The legend details for the X and Y axis of the Figure 10 has been added and the figure has been modified accordingly.
Comment#9: Figure 12: you need to describe in the legend what cisplatin is and why did you use that. Also this reagents needs to be included and described in the materials and methods sections. You need also to include a non-treated control in figure 12a like what you did in figure 12b.
Response: We have added control without treatment flow cytometric analysis in the Figure 12a.
Comment#10: line 522: it is known that there is potential predation of mosquito larvae by copepods. the question and the potential contribution of your work is if the presence of silver NPs facilitate this predation.
Response: When there were silver nanoparticles, the mosquito movement was slowed down and at that time, the predation rate/percentage was high. Moreover, the physiological status of the larvae was disturbed. This explanation was mentioned in the respective section of the revised version.
Comment#11:line 523: efficiency of what?
Response: Predatory potential/ No. of predation/ ability of predation and mentioned in the text.
Comment#12: line 561-563: you need to rewrite this part. line 561, the most effective is too ambitious. There are several ways to eliminate mosquitos. You just showed that the presence of NPs slightly improved the action of copepods. line 562-563 I can't follow that and I don't understand this conclusion from the authors.
Response: The conclusion part was simplified and mentioned in the respective section with yellow highlighted text.
Comment#13: Finally a major issue is related to the scratch migration assay. the authors should provide a control and then compare the healing with or without NPs and include the control as extra panels in Figure 11 for 0, 12, and 24 hour. As soon as you include the control and after comparing the images with NPs you can then say what you claim in lines 489-490 about the migratory capacity of the carcinoma cells.
Response: We have added a control untreated panel of cells in the scratch migration assay figure of the revised version.
END

Round 2
Reviewer 2 Report
Author have addressed the issues raised throughout the first round of reviews.